# Unveiling the Role of Human Papillomavirus in Urogenital Carcinogenesis a Comprehensive Review

**DOI:** 10.3390/v16050667

**Published:** 2024-04-25

**Authors:** Beliz Bahar Karaoğlan, Yüksel Ürün

**Affiliations:** 1Department of Medical Oncology, Ankara University Faculty of Medicine, 06620 Ankara, Türkiye; bbaharulas@gmail.com; 2Faculty of Medicine, Department of Internal Medicine, Division of Internal Medicine, Ankara University Cancer Research Institute, 06620 Ankara, Türkiye

**Keywords:** anal cancer, bladder cancer, HPV, Human papillomavirus, kidney cancer, male health, oncogenic virus, penile cancer, prostate cancer, renal cell carcinoma, testicular cancer, urogenital cancers, vaccine

## Abstract

Human papillomavirus (HPV), an oncogenic DNA virus, is the most common sexually transmitted virus and significant public health concern globally. Despite the substantial prevalence of HPV infection among men, routine testing remains elusive due to the lack of approved HPV tests and the complexity of detection methods. Various studies have explored the link between HPV and genitourinary cancers, revealing different associations influenced by geographic variation, histological subtype and methodological differences. These findings underscore the importance of further research to elucidate the role of HPV in male urogenital cancers. This comprehensive review delves into the intricate relationship between HPV and male genitourinary cancers, shedding light on the virus’s oncogenic mechanisms and its reported prevalence. A deeper understanding of HPV’s implications for male health is essential for advancing public health initiatives and reducing the burden of urogenital cancers worldwide.

## 1. Introduction

Human papillomavirus (HPV) is an oncogenic virus that primarily infects epithelium or mucous membranes. Mucosal HPVs, transmitted through sexual contact, is the most common sexually transmitted infection worldwide [1]. The prevalence of genital HPV infection among men is significant, with approximately one-third of men worldwide being infected by at least one type of genital HPV [2]. High-risk HPV types can lead to precancerous lesions and are responsible for about 10% of cancers globally, including over 90% of cervical cancers, most anal cancers, and some vulvar, vaginal, penile, and head and neck cancers [1]^.^ HPV infection can persist for years before progressing to invasive cancer. Understanding these mechanisms is crucial for developing effective treatments and preventive measures against HPV-related cancers.

In recent years, there has been growing interest in the high prevalence of HPV in men and its potential role in the etiology of urogenital cancers. There are two hypotheses regarding the link between HPV and male genitourinary cancers. The first suggests an anatomical basis, as the urethra serves as a reservoir and direct link between the urinary and genital area, providing a natural route for viral migration. The second hypothesis pertains to the natural affinity of HPV for epithelial cells. Typically, these viruses infect epithelial cells with a strong preference for squamous epithelium, indicating a propensity for urethral and bladder epithelium as well [3]. The aim of this study is to elucidate the potential role of HPV in the etiology of male urogenital cancers, by examining its oncogenic effects and reported frequency, within the context of current scientific evidence.

## 2. HPV Types and Oncogenesis

HPV-DNA is a circular, double-stranded molecule surrounded by a protein coat. It contains eight genes, split into early (E) and late (L) stages. E1 and E2 handle replication, transcription, and cell regulation. E4 assists in cell cycle control and virion assembly. E5 controls cell growth and virus modulation. E6 and E7 inhibit apoptosis and regulate cell cycling.

HPV types fall into two categories: cutaneous and mucosal. Mucosal HPVs, transmitted through sexual contact, include high- and low-risk oncogenic strains. Low-risk types like 6 and 11 cause genital warts and recurrent respiratory papillomatosis, which seldom progress to cancer. In contrast, persistent infection with high-risk types like 16, 18, 31, 33, 45, 52, and 58 can lead to precancerous lesions and are responsible for about 10% of cancers globally, including over 90% of cervical cancers, most anal cancers, and some vulvar, vaginal, penile and head and neck cancers [1].

## 3. Human Papillomavirus (HPV) and Its Role in Carcinogenesis

The most well-known mechanism of HPV in cancer development involves its oncoproteins E6 and E7 (Figure 1).

When HPV DNA integrates into the host genome, E6 and E7 inactivate tumor suppressor proteins such as p53 and Rb. High-risk HPV E6 proteins can activate telomerase, a key enzyme for maintaining telomere length and ensuring cell immortality [4]. Additionally, E6 can hinder DNA repair by binding and inhibiting DNA repair proteins, leading to an accumulation of mutations in the host genome [5]. Meanwhile, the E7 protein can disrupt centrosome duplication by affecting cyclin E/cyclin-dependent kinase 2 (CDK2) complexes, causing genomic instability and aneuploidy [6]. All these proccess leads to increased cell proliferation, apoptosis resistance, and malignant transformation. Additionally, HPV triggers chronic inflammation in prostate tissue, contributing to the oncogenic process [7,8,9]. Another mechanism involves HPV disrupting the activity of the protective enzyme APOBEC3B against viral infections. It’s believed that this process resulting in genomic instability and oncogenesis is due to the indirect effect of HPV [10,11]. Persistent infection with high-risk HPVs and progression from latent infection to invasive cancer can take years to decades [4,12].

## 4. Immune Response to HPV Infection

Host defense mechanisms effectively control initial HPV infections in most individuals, with only a subset progressing to invasive cancer. Physical barriers like mucous membranes blocks virus entry, virus innate immunity recognizes HPV DNA through pathogen sensors. HPV alters host gene expression to evade virus responses by deregulating DNA methylation, histone modification, and nuclear factor kappa B (NF-κB) signaling. The virus interacts with host proteins to inhibit virus responses, impairing antigen presentation and promoting virus suppression. HPV manipulates protein functions to evade virus responses by disrupting protein-protein interactions, inhibiting IFN signaling, and inducing proteasome-mediated degradation of cytokines. The virus also interferes with MHC molecule trafficking and antigen presentation [11].

## 5. Detection of HPV

Determining the link between exposure and outcome relies on accurately identifying the exposure. Various methods are used in different studies to detect HPV, each with its own strengths and limitations.

While the role of HPV in women’s health is well understood and extensively studied through various screening and research programs, men have been significantly less involved in these efforts. This is partly because HPV-related diseases affect and result in fatalities among women more frequently than men [13]. Although there’s been increased focus on HPV infection in males in recent years, awareness and understanding of HPV-related issues in men still lag behind that in women. Currently, there isn’t a universally accepted and validated test for screening HPV in males [14]. The lack of routine screening for HPV in men can be attributed to inconsistent results observed in testing. Obtaining samples from penile skin poses challenges due to its anatomical structure, which is less permeable compared to that of women. As a result, the efficacy of HPV testing methods in men remains uncertain.

While obtaining samples for HPV detection in men isn’t as straightforward as cervical smear collection, optimal sampling site for HPV testing in men remains unknown. Penoscopy may be useful for detecting warts and carriers but has limited sensitivity for flat lesions. Urethroscopy is recommended only for condylomas in the urethral meatus [15]. Other methods include investigating HPV presence in body fluids like urine and semen. Serological methods that detect antibodies against HPV in serum have low sensitivity due to the virus’s weak immune response in many cases and may not provide information about infection activity or localization [16].

Morphological changes such as papillomatosis, hypergranulosis, acanthosis, and koilocytosis in cancerous tissues indicate HPV infection but with low sensitivity. Immunohistochemical (IHC) methods for detecting HPV capsid proteins are highly specific but have low sensitivity. Over the past decades, there has been significant advancement in the methods used to detect HPV infection. Earlier techniques like hybridization/blotting have been replaced by more efficient signal amplification assays, particularly quantitative polymerase chain reaction (qPCR), which can identify individual HPV genotypes. These qPCR tests have become the primary approach for HPV screening and clinical diagnosis [17,18].

Next-generation sequencing (NGS) offers a highly sensitive method for HPV detection, capable of identifying low-copy-number types, novel variants, and even those that may evade detection by standard molecular methods. Whole-genome NGS, covering the entire HPV genome, allows for precise identification of variants and subvariants beyond genotype level.

Recent studies utilizing NGS have provided more accurate estimates of HPV’s contribution to cervical cancer compared to traditional PCR techniques [19,20,21]. Discrepancies in HPV testing methodologies across countries and studies primarily arise from differences in sample types and the scientific resources at their disposal. This situation often leads to methodological inconsistencies in studies aiming to evaluate the relationship between HPV and organ-associated cancers, thereby compromising the reliability of the data. With NGS technology becoming more cost-effective and robust, it is poised to rival existing HPV-DNA tests. Furthermore, NGS holds promise in elucidating the genotype-phenotype relationships in HPV-associated malignancies such as male urogenital cancers, offering insights into the underlying molecular mechanisms of these diseases.

## 6. Penile Cancer

Penile cancer, with an average annual incidence of 1 per 100,000 men, is a rare malignancy [22]. Despite its rarity and the curative potential of early-stage surgical intervention, the physical and psychosocial morbidity associated with the disease and its treatment is significant. Etiological factors are HPV infection, poor hygiene, lack of circumcision, inflammatory conditions, lichen sclerosis, immunosuppression, and smoking [23]. It typically arises from mucosal surfaces and the most common localizations are glans penis and inner prepuce [24]. Squamous cell cancer (SCC) is the most common type and carious subtypes of SCC exist, including usual, papillary, condylomatous (warty), basaloid, verrucous, and sarcomatoid [25,26]. Penile SCC mainly impacts older men, with the highest incidence occurring in their sixties [27]. Recent observations suggest a rise in cases among younger individuals, possibly linked to changes in sexual behaviors leading to increased exposure to sexually transmitted diseases and higher rates of HPV infection [28].

Studies suggest approximately 40% of penile cancers are HPV-positive [29,30]. In a retrospective analysis of over 1200 penile cancer cases by Backes et al., HPV positivity was observed in 48% of patients, with the highest risk of HPV-related penile cancer development reported in Asia [31]. HPV types 16 and 18 are the most commonly detected in penile cancer cases, with both types present together in about one-third of cases [30,32,33]. The prevalence of HPV positivity varies among different SCC subtypes, with higher rates observed in basaloid and condylomatous types [31,34].

In penile intraepithelial neoplasia (PeIN), known as a precursor lesion of penile SCC, HPV positivity rate increases with higher grades of dysplasia [30]. PeINs are now classified as HPV-independent or HPV-associated according to the 2022 World Health Organization (WHO) classification [35].

Uncircumcised men have higher HPV prevalence and it seem like circumcision protects against HPV and HPV-related penile cancer [27,36]. The prognostic significance of HPV in penile cancer, which is well-established in head and neck cancers [37], remains an area of research interest. Studies suggest that HPV presence is associated with a favorable prognosis in penile cancer [38,39], while a few suggests a poorer prognosis [40]. A better understanding of HPV-related oncogenic mechanisms will likely shed light on the prognostic significance of HPV in penile cancer.

In conclusion, penile cancer, though rare, warranting attention to preventive measures and early detection strategies. HPV infection, among other factors, plays a pivotal role in its etiology, with HPV-positive cases exhibiting varying prognostic implications.

## 7. Anal Cancer

Anal cancer is a relatively rare malignancy, but its incidence has been rising steadily over the past few decades [22]. Anal squamous cell carcinoma (ASCC) is the most common histologic type. HPV infection has emerged as a significant risk factor for the development of ASCC. High-risk HPV has been detected in 90% of ASCCs, with HPV-16 is the most common type [41,42]. Anal intercourse and a high number of sexual partners throughout life elevate the risk of persistent HPV infection, leading to subsequent high-grade squamous intraepithelial neoplasia (HSIL) and ASCC. Additionally, factors such as immunosuppression resulting from HIV infection or the use of immunosuppressants post-solid organ transplantation, hematologic malignancies, prior HPV-related cancers, autoimmune disorders, low socio-economic status, and smoking history are also associated with an increased risk of ASCC [43].

ASCC is considered to share similarities in its biology and natural history with cervical cancer, including association with high-risk HPV infection to precancerous lesions to cancer. Similar to the cervical transformation zone, the virus targets actively dividing basal cells located in the transition zone, situated within the rectal columnar mucosa distal to the dentate line, and spreads towards the squamocolumnar junction. Anal Papanicolaou (Pap) smears involve cytological examination beyond the squamocolumnar junction but this method lacks utility as a screening tool in high-risk populations. The gold standard is high-resolution anoscopy (HRA), which involves examining the squamocolumnar junction, anal canal, and perianal skin under magnification using an anoscope [44].

Anal HPV prevalence varies substantially by HIV-status and sexual orientation in men, which is considered to increase HPV prevalence at the anal site [45]. A recent systematic review, encompassing a large cohort of nearly thirty thousand men, assessed the prevalence of anal HPV and HSIL, stratified by HIV status and sexual orientation. Among HIV-negative men who have sex with women (MSW), the prevalence of anal HR-HPV was found to be at 6.9%. In contrast, among HIV-positive MSW, the prevalence rates were notably higher at 26.9% for HR-HPV. Among HIV-negative MSM, the prevalence was 41.2%, while among HIV-positive MSM, the prevalence rates were substantially elevated at 74.3% [46].

Findings from the U.S. Anal Cancer HSIL Outcomes Research (ANCHOR) study revealed that treating anal precancers notably diminishes the risk of progressing to anal cancer, particularly among people living with HIV aged 35 years and older [47]. These outcomes emphasize the urgency of identifying effective methods for detecting anal precancers that can be promptly treated to avert further progression. Guidelines recommend the implementation of screening programs utilizing anal cytology and high-resolution anoscopy for high-risk populations, such as gay, bisexual, and other MSM, as well as HIV-negative women with a history of anal intercourse or other HPV-related anogenital malignancies. These recommendations draw inspiration from the successes observed in cervical cytology screening. However, the absence of randomized controlled studies demonstrating the preventive efficacy of screening in these high-risk populations precludes its routine endorsement at present [43].

The high prevalence of anal HPV among young MSM underscores the importance of gender-neutral HPV vaccination prior to sexual debut, compared to catch-up vaccination efforts. Additionally, HIV-positive MSM represent a priority group for ASCC screening initiatives [46]. Consistent with these findings, another study investigating the efficacy of HPV vaccines in preventing anal precancerous lesions and ASCC revealed compelling evidence supporting the substantial effectiveness of vaccination in reducing anal HPV infection and HSIL among HIV-negative individuals vaccinated at or before the age of 26. Importantly, the limited impact observed in individuals HIV-positive beyond this age suggests that vaccines may confer greater benefits in populations with lower levels of sexual exposure to anal HPV [48].

The presence of HPV infection significantly impacts the prognosis of ASCC. Individuals with HPV-negative tumors are less likely to respond to CRT than those with HPV-positive tumors [49,50]. A meta-analysis has shown that patients with HPV-positive/p16-positive tumors have improved survival outcomes compared with patients with either HPV-negative/p16-positive or HPV-positive/p16-negative tumors [51].

In an NGS study of HPV in ASCC, samples with viral integration had higher PIK3CA-activating mutation rates, which is an APOBEC editing signature in HPV-positive HNSCC were associated with increased mutational burden, suggesting immunogenicity and suitability for immune therapy [52].

In conclusion, the escalating incidence of anal cancer, primarily attributable to HPV infection, necessitates a multifaceted approach for effective management. While HPV vaccination represents a crucial preventive strategy, screening programs utilizing anal cytology and HPV testing hold promise, particularly for high-risk groups.

## 8. Prostate Cancer

Prostate cancer (PCa) is the most common solid tumor in men and ranks high among cancer-related deaths [53]. Alongside known factors like age, family history, and genetic risk; modifiable risk factors such as smoking, diet, obesity, and reduced physical activity play a role in PCa etiology [54,55,56].

Prostate tissue is predominantly glandular in origin, it also contains a transition zone similar to the cervix. However, only about a quarter of PCa cases originate from this zone [57]. Although squamous metaplasia can occasionally occur in PCa tissues, it is known to be hormonally driven. Glandular structures are dominant in PCa [58,59,60]. While it could be speculated that squamous metaplasia observed in the prostate may be HPV-related, it could also serve as a potential reservoir for HPV.

HPV presence in prostate tissue was first identified by McNicol and Dodd using PCR method [61]. Subsequent studies showing koilocytosis in PCa tissues suggest a potential role for HPV in PCa etiology [62,63]. Most studies investigating the relationship between HPV and PCa use tissues from patients with benign prostatic hyperplasia (BPH) as the control group [63,64,65,66,67]. While understandable due to the quantitative advantage over normal prostate tissues, this approach can lead to confusion considering that pre-cancerous lesions are often evaluated under the umbrella of BPH. A meta-analysis, conducted by Tsydenova IA et al., included 27 studies and 3122 tissue samples, focusing solely on studies using PCR-based methods, showed a significant association between HPV and PCa in normal tissues, although among the studies using BPH tissues as controls, there was no significant relationship between HPV presence and PCa. These results highlighted how even the choice of control tissue can influence outcomes [68].

The most comprehensive review examining the relationship between PCa and HPV included 60 studies, with 51 studies investigating HPV presence in tissues and 13 studies in blood, using various methods including PCR, ELISA, hybridization, and IHC. Among the included studies, 11 (18%) showed a positive association between HPV presence and PCa development [69].

Research exploring the potential role of HPV in PCa pathogenesis has utilized various methodologies, yet findings remain inconclusive. Traditional PCR-based techniques, commonly employed in prior studies, may lack the sensitivity required to detect past HPV infections, limiting their ability to elucidate the HPV-PCa relationship [63,64,65,66,67,70]. In contrast, a study by Khoury et al., utilizing NGS methods on the Human Genome Atlas database, found no HPV presence in 53 PCa cases [71]. Given the high sensitivity and specificity of the method employed, this study stands out with a high level of evidence compared to many others with different methodologies. Further studies using similar methodology are crucial to investigate the relationship between HPV and Pca.

The timing of human papillomavirus (HPV) involvement in the oncogenic process is another unresolved problem. Although HPV is implicated in both cervical cancer formation and progression, its role in PCa is thought to involve a ‘hit and run’ phenomenon, wherein HPV triggers early malignant transformation in prostate cells and then becomes cleared, explaining the absence of viral genome detection in PCa tissues [72,73,74,75,76]. Some studies propose HPV’s effectiveness in later stages of oncogenesis, associating HPV presence with a high Gleason score, contradicting the ‘hit and run’ hypothesis [65]. However, many studies have not confirmed this relationship [72,77]. These inconsistencies underscore the need for further research to elucidate the complex interplay between HPV and PCa development. Additionally, the potential impact of HPV vaccination on PCa risk remains uncertain, warranting future studies to explore preventive measures in PCa carcinogenesis.

In conclusion, existing studies on the association between HPV and PCa are hindered by methodological variations and diverse control groups, resulting in insufficient evidence to conclusively establish the role of HPV in PCa etiology. Despite suggestive findings, the current body of literature lacks robustness, emphasizing the inadequacy of available data to firmly support the involvement of HPV in PCa carcinogenesis. Conflicting results and differing interpretations underscore the necessity for more comprehensive investigations in this field. Moreover, as all studies to date have been observational in nature, the potential impact of HPV vaccination on PCa risk remains uncertain and warrants further exploration.

## 9. Bladder Cancer

Bladder cancer (BC) ranks as the tenth most common cancer globally [53]. It originates from the urothelium, which is a waterproof protective barrier against urinary tract infections. It undergoes rapid shedding and regeneration in response to acute injury or infection.

Histologically, BC is categorized into different types, including urothelial carcinoma (UC), squamous-cell carcinoma (SCC), and adenocarcinoma. UC constitutes about 95% of BC cases, encompassing various differentiated and histologic subtypes. BC can be further classified based on invasion depth into muscle-infiltrating BC (MIBC) and non-muscle-infiltrating BC (NMIBC). NMIBC affects around 75% of patients [78].

Smoking remains a significant independent risk factor for BC development, with persistent smokers having a threefold higher risk compared to non-smokers. Schistosomiasis is recognized as a notable cause of SCC. Other factors contributing to BC occurrence include exposure to occupational carcinogens, genetic predisposition, and dietary habits [79]. Lately, there’s been a focus on the link between BC and infections like HPV and similar viruses [80,81,82].

Numerous investigations have explored how HPV infection impacts the bladder epithelium. Furthermore, HPV DNA has been identified in urine and washing specimens from individuals with BC, along with BC tissues [83]. Additionally, there’s a positive correlation between high-risk HPV infection and tumor advancement, with the latter often linked to p53 mutation [84].

In a meta-analysis conducted by Sun J. et al., comprising 6065 BC patients, the prevalence of HPV was found to be 16% (11–21%). Studies indicated that HPV increases the risk of BC (OR: 3.35, 95% CI: 1.75–6.43) and its recurrence (OR:1.87, 95% CI: 1.24–2.82), which showed the latent prognostic effect of the virus [85]. In a meta-analysis conducted by Muresu et al., a HPV prevalence of 19% was reported. HPV types 16 and 18 were most commonly detected in BC samples, and HPV was shown to increase the risk of BC by more than sevenfold (OR: 7.84, 95% CI: 4.34–14.15) [86]. Studies have also found that the virus is associated with muscle invasion, high grade, and advanced stage disease in patients with BC [87,88]. However, these results were influenced by heterogeneous variables such as region, the method used to detect HPV (PCR vs. others), sample (e.g., FFPE, fresh tissue), and BC histology.

Asia has the highest prevalence of HPV, with reported HPV positivity reaching up to 45% in studies [85]. The difference in prevalence between continents may be attributed to factors such as genetic differences, cultural disparities, and sexual habits, in addition to the method used to detect HPV. Furthermore, analysis of studies conducted in Asia indicates that the presence of HPV increases the risk of BC by more than sixfold (OR: 6.2, 95% CI: 2.167–18.250) [89].

The relationship between HPV and BC varies according to the histological subtypes. Due to the predominant histological subtype being UC, many studies have indicated an association between HPV and UC, while data showing an increased risk for SCC are limited [58,59]. Interestingly, in patients with UC, squamous differentiation associated with HPV has been shown to negatively affect prognosis [90,91].

While the exact connection between HPV infection and the development and advancement of BC isn’t fully understood, a comprehensive review of existing literature indicates that there is a positive link between HPV infection (especially high-risk types) and BC, particularly UC, although some studies suggest that HPV may not be a direct cause [92,93,94].

## 10. Renal Cell Carcinoma

Renal cell carcinoma (RCC) is one of the common cancers in men, with an incidence rate of 4.4% reported worldwide [95]. The predominant histological type, clear cell RCC, accounts for the majority of cases. Risk factors for RCC are advanced age, male gender [96], genetic factors (e.g., von Hippel-Lindau syndrome) [97,98], smoking [99], diet, hypertension [100], obesity [101], and environmental exposures (e.g., cadmium and asbestos) [102,103].

The potential role of HPV in RCC etiology was first studied in the early 1990s [104]. Subsequent research has explored whether this virus, commonly found in the male genital system, contributes to RCC risk. In a study by Farhadi et al., HPV was detected in 30% of tumor tissues from RCC patients. The absence of HPV’s L1 capsid protein and cellular changes associated with the virus, such as koilocytosis, in RCC patients led to the interpretation that HPV’s life cycle is disrupted in advanced stages. The presence of HPV was also associated with high-grade tumors in this study [105]. Another study showed a 14.3% prevalence of HR-HPV in RCC samples using PCR, while HPV was not found in any healthy renal tissue in the control group, suggesting a potential role for HPV in RCC etiology [106]. Additionally, a study using ISH demonstrated a 52% positivity rate for HPV-DNA in RCC tissues [107]. However, a comprehensive study by Khoury et al. using data from the Human Genome Atlas did not establish a relationship between RCC and HPV [71]. Similarly, another study involving various histological types of RCC found no association between HPV and RCC [108]. Apart from case reports and case-control studies, there is a lack of extensive research on this topic. Given the heterogeneity of methods used to detect HPV-DNA and the conflicting results of studies, it is not possible to definitively conclude that HPV plays a role in RCC etiology based on current evidence. Further large-scale studies using standardized methods are needed to investigate this relationship.

## 11. Testicular Cancer

Testicular cancer (TC) is the most common solid tumor in men aged 20–40 years, comprising approximately 1–2% of all cancers. It is histologically classified into germ cell tumors (TGCT) and non-germ cell tumors. Risk factors for TC include cryptorchidism, genetic predisposition, and substance use [109,110]. The similarity in age of onset between TC and diseases caused by sexually transmitted infections raises the possibility of infectious agents playing a role in the etiology of TC.

In a case-control study investigating HPV presence in semen samples using PCR and FISH, HPV was detected in 9.7% of patients diagnosed with TGCT and 2.4% of healthy controls [111]. Another study by Stricker et al. evaluated HPV antibody levels in 39 TC patients using ELISA, revealing a 5% HPV positivity rate, although HPV presence in malignant tissues of these patients was not examined [112]. This finding is not sufficient to demonstrate an association between HPV presence and TC. Indeed, studies evaluating HPV in tissues using PCR have not found HPV presence in either patients or control groups [113,114].

Recent studies have highlighted the potential role of HPV in male infertility by directly infecting male gametes, leading to decreased fertility due to increased sperm DNA damage and abnormal chromosome numbers. HPV is believed to attach to sperm cells at specific sites on their heads, similar to other viruses that infect sperm [115,116]. With increasing evidence suggesting a link between HPV and male infertility, questions have arisen about its role in TC. But when assessing the correlation between testicular germ cell neoplasms and HPV, it is crucial to consider that HPV exhibits a predilection for epithelial tissues, whereas testicular germ cell neoplasms originate from non-epithelial sources. Further research of high quality is required to better understand this relationship.

## 12. HPV Vaccination

The optimal long-term approach for mitigating the risks associated with HPV-related cancers is vaccination. Vaccines consist of virus-like particles (VLPs) designed to elicit immunity against specific HPV types included in the vaccine formulation. The initial efficacy trials focused on quadrivalent (4vHPV) vaccine, containing HPV-16, HPV-18, HPV-6, and HPV-11 VLPs and bivalent vaccine (2vHPV), containing HPV-16, HPV-18, HPV-6, and HPV-11 VLPs. There is also nine-valent HPV vaccine (9vHPV), provides additional protection against HPV-31, 33, 45, 52, and 58. The Centers for Disease Control (CDC) recommends HPV vaccination for individuals aged 11 to 12 years, with females up to the age of 26 years, boys up to the age of 21 years, and specific populations up to the age of 26 years recommended for catch-up immunization [117]. 9vHPV vaccine is also indicated for preventing anogenital lesions in both men and women up to the age of 45, as well as oropharyngeal and other head and neck cancers caused by select HPV types [117].

HPV vaccination elicits robust levels of HPV-type-specific antibodies, particularly in children under 16 years, compared to older age groups [118]. Despite the demonstrated efficacy of HPV vaccination in males, several countries have yet to implement routine vaccination for boys, resulting in notably low vaccination coverage among males. Efforts to enhance medical awareness regarding the benefits of male HPV vaccination may enhance disease management and increase vaccine acceptance among parents and boys. A systematic review assessing HPV vaccination efficacy in male populations found significant effectiveness against HPV-related anogenital diseases, with the highest efficacy observed against anal intraepithelial neoplasia (AIN) and genital condyloma [119].

The era of prophylactic vaccination underscores the importance of updating the potential individual benefits for boys receiving the HPV vaccine, especially in light of the rising frequency of head and neck HPV-associated cancers. Lechner et al. highlighted key points indicating that the incidence of HPV-associated oropharyngeal cancer is expected to rise until the benefits of gender-neutral prophylactic HPV vaccination become evident [120]. Also, individuals living with HIV and MSM face elevated risks of HPV-related diseases due to HIV-induced immunosuppression. Although vaccinating women can confer indirect protection to heterosexual men, MSM and bisexual men do not benefit from this herd immunity [119].

Although the existing vaccines are primarily preventive, they have shown advantages in individuals undergoing treatment for HPV-related conditions such as cervical intraepithelial neoplasia (CIN), genital warts, anal neoplasia, and recurrent respiratory papillomatosis (RRP). The mechanism behind inhibiting further disease progression is believed to involve halting the spread of HPV [121,122].

Another area of research pertains to the protective efficacy of single-dose HPV vaccination. In a randomized, multicenter, double-blind, controlled trial conducted in Kenya in 2022, involving women aged 15–20 years, single-dose administration of 9vHPV or 2vHPV vaccines revealed that a single dose effectively prevented incident persistent oncogenic HPV infection, demonstrating comparable efficacy to multidose regimens assessed over an 18-month period [122]. However, evidence regarding the effectiveness of a single dose of HPV vaccine in boys is currently lacking.

Young cancer survivors have a higher risk of developing HPV-related malignancies compared to the general population, often due to underimmunization [123,124]. HPV vaccines are effective in these individuals, with timing based on individual risk factors and no need for prior HPV testing [125].

## 13. Conclusions and Future Directions

Unraveling the complex interplay between HPV and genitourinary cancers in men requires further research. While evidence suggests a potential link, inconsistencies and methodological limitations necessitate a multifaceted approach (Figure 2). Standardizing HPV detection methods across healthcare settings is essential to ensure consistency and reliability in diagnosis. Longitudinal studies are needed to track HPV progression over time and understand its evolution into cancer.

Evaluating the impact of HPV vaccination on cancer incidence will provide valuable insights into the effectiveness of vaccination programs. Deciphering the oncogenic mechanisms of HPV in genitourinary tissues is crucial for identifying potential therapeutic targets. Identifying high-risk populations and developing targeted interventions can help tailor prevention and screening strategies. Additionally, developing specific diagnostic tools for early detection and conducting comprehensive epidemiological surveys are essential for global mapping and informed public health policies. Promoting vaccination and safe sex practices through targeted public health campaigns can increase vaccine uptake and reduce HPV transmission. Evaluating the feasibility and benefits of routine HPV testing in clinical settings is necessary for effective screening programs. Lastly, investigating the efficacy of existing and novel therapies for HPV-positive cancers will improve treatment outcomes and patient survival. By prioritizing these research avenues, we can enhance our understanding of the HPV-genitourinary cancer link and achieve significant advancements in public health.

## Figures and Tables

**Figure 1 viruses-16-00667-f001:**
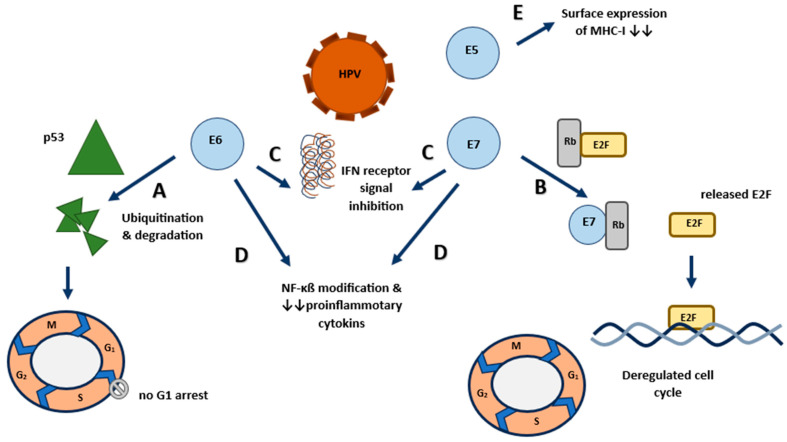
Effects of HPV on host cells. HPV disrupts cell cycle regulation post-infection through two main mechanisms: (A) Firstly, the E6 oncoprotein binds to P53, triggering proteolysis via the ubiquitination pathway, activating the CDK cycle and cell cycle. (B) Secondly, the E7 oncoprotein binds to Rb and other associated proteins, releasing the E2F transcription factor complex and activating the cell cycle. (C) HPV inhibits type 1 IFN-γ receptor with E6 and E7 oncoproteins, and it inhibits type 2 IFN-γ receptor signaling with E7 oncoproteins. (D) E6 and E7 oncoproteins also inhibit NF-κB signaling. (E) E5 and E7 prevent surface expression of MHC molecules. E5 binds to MHC-I and MHC-II in the Golgi and ER, inhibiting their transport to the cell surface. Abbrevations: CDK: Cyclin-dependent kinase, ER: Endoplasmic reticulum, HPV: Human papillomavirus, IFN-γ: interferon-gamma, NF-κB: Nuclear Factor kappa B.

**Figure 2 viruses-16-00667-f002:**
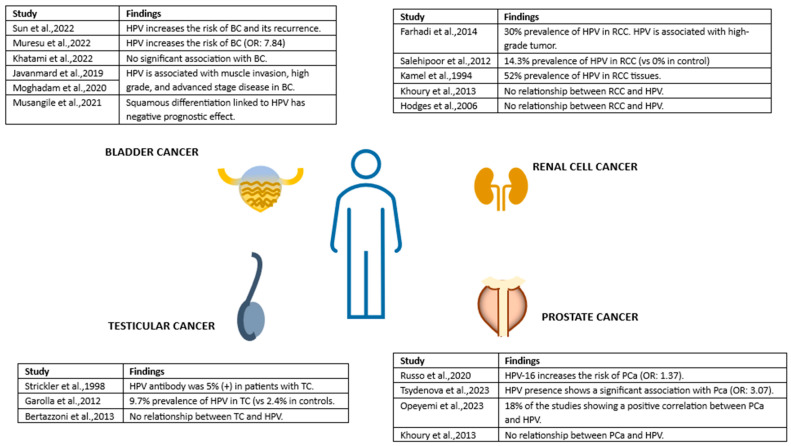
Unraveling Possible Male Urogenital Cancers Linked to HPV: Summary of findings. References cited in the figure: [85,86,87,88,89,91] for BC, [71,105,106,107,108] for RCC, [111,112,114] for TC and [68,69,70,71] for PCa. Abbrevations: BC: Bladder cancer, HPV: Human papillomavirus, PCa: Prostate cancer, RCC: Renal cell carcinoma, TC: Testicular cancer.

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
