# Peer review of "Unveiling the Role of Human Papillomavirus in Urogenital Carcinogenesis a Comprehensive Review"

_viruses, 2024, doi:10.3390/v16050667_

Round 1

Reviewer 1 Report

Comments and Suggestions for Authors

 In this review, KaraoÄŸlan and Ürün described the role of HPV in a set of urogenital malignancies (prostate, bladder, renal cell, and testicular carcinomas). The manuscript is comprehensively presented, highlighting 83  articles, including meta-analyses. A few points should be clarified and or justified in the manuscript:

1. It is unclear why penile cancer was excluded, considering that the literature demonstrated the role of HPV in the risk of these tumor types.

2. I suggest a new topic with more details about HPV infection in cancer patients and vaccination. 

3. item  11, "Future Directions"  should be revised. The authors should replace the items presented in "numbers" with a more elaborate paragraph.

Minor comments: 

Line 102: complete the meaning of the abbreviation CDC (Centers for Disease Control and Prevention).

Lines 102-103: Please explain better why HPV routine testing is not recommended due to the high prevalence of infection. In my opinion, this is not the case.

Lines 106-107: "optimal sampling methods for HPV detection in men remain unknown" or the definition of the more adequate location to collect the sample?  

Lines 201-202: the sentence is repetitive: "This contradicts the 'hit and run' phenomenon suggesting that the virus only plays a role in the early stages of PCa development[27]"

Comments on the Quality of English Language

Adequate.

Author Response

Reviewer 1

In this review, KaraoÄŸlan and Ürün described the role of HPV in a set of urogenital malignancies (prostate, bladder, renal cell, and testicular carcinomas). The manuscript is comprehensively presented, highlighting 83  articles, including meta-analyses. A few points should be clarified and or justified in the manuscript:

  1. It is unclear why penile cancer was excluded, considering that the literature demonstrated the role of HPV in the risk of these tumor types.

*** Sections on the relationship between HPV and anal and penile cancer have been added to the manuscript.

  1. I suggest a new topic with more details about HPV infection in cancer patients and vaccination. 

*** A new section on HPV&vaccination, including cancer patients, has been added to the manuscript.

  1. item  11, "Future Directions"  should be revised. The authors should replace the items presented in "numbers" with a more elaborate paragraph.

*** Revised accordingly.

Minor comments: 

Line 102: complete the meaning of the abbreviation CDC (Centers for Disease Control and Prevention).

** This section has been revised. The revised sentences are highlighted.

Lines 102-103: Please explain better why HPV routine testing is not recommended due to the high prevalence of infection. In my opinion, this is not the case.

** Explained as “The lack of routine screening for HPV in men can be attributed to inconsistent results observed in testing. Obtaining samples from penile skin poses challenges due to its anatomical structure, which is less permeable compared to that of women. As a result, the efficacy of HPV testing methods in men remains uncertain.”

Lines 106-107: "optimal sampling methods for HPV detection in men remain unknown" or the definition of the more adequate location to collect the sample?  

** The phrase "The optimal sampling site for HPV testing" has been replaced.

Lines 201-202: the sentence is repetitive: "This contradicts the 'hit and run' phenomenon suggesting that the virus only plays a role in the early stages of PCa development[27]

** This section has been revised. The revised sentences are highlighted.

Reviewer 2 Report

Comments and Suggestions for Authors

The paper entitled “Unveiling the Role of Human Papillomavirus in Urogenital Carcinogenesis: A Comprehensive Review” by KaraoÄŸlan et al. is a review of the literature focused on HPV detection in renal, bladder, prostate and male gonads. The authors underline that literature data are controversial and conclude that “Unraveling the complex interplay between HPV and genitourinary cancers in men (?) requires further research”.

The paper is poorly informative since it does not establish a critical hierarchy between the different methodology used in the respective papers. As an example, in the important paragraph on prostate cancers, the paper by Khoury et al., based on a powerful NGS approach, reports that the detection of HPV sequences with extensive molecular characterization (integration loci etc.) was positive in several types of tumors but that all of the prostate cancer specimens anayzed remained negative. This very strong paper is on the same level as others based on less specific approaches. In other words, if more stringent criteria were used for the analysis of the methodology, the conclusion would be very different and, consequently, the review much more useful for the readers. As a possible approach for instance, this review could highlight HPV detection performed using up to date NGS-based approach, and include anal cancers in their review (may be a review focused on male patients as suggested in the conclusion?). The comparison between the robustness of the data provided by NGS-based investigations on anal carcinoma (Morel, Cancers, 2019) and the paucity of those obtained by the analyses, using similar approaches, of tumors developed in other organs (Khoury for prostate, others most probably exist for bladder, …) should be striking. The authors might thus conclude that HPVs are oncogenic on the anal mucosa (which presents a transition zone) but that the data available on tumors developed in other organs point rather towards passenger infections or mere risk factors of HPVs with no evidence of direct oncogenic effect.

At the era of prophylactic vaccination, it would be important to update the potential individual benefit for boys receiving the HPV vaccine. Of note, there is a strong increase in frequency of head & neck HPV-associated cancers (Lechner, Nature, 2022). A mention of this might also be included, providing a global overview of HPV-associated cancers in males, and reinforcing the quality of the paper in the perspective of vaccine against HPVs in males.

Comments on the Quality of English Language

Correct presentation

Author Response

Point-by-point Response to Reviewers

Reviewer 2

The paper entitled “Unveiling the Role of Human Papillomavirus in Urogenital Carcinogenesis: A Comprehensive Review” by KaraoÄŸlan et al. is a review of the literature focused on HPV detection in renal, bladder, prostate and male gonads. The authors underline that literature data are controversial and conclude that “Unraveling the complex interplay between HPV and genitourinary cancers in men (?) requires further research”.

** The title has been appropriately revised. Additionally, the sections on penile and anal cancers have been added to the review.

The paper is poorly informative since it does not establish a critical hierarchy between the different methodology used in the respective papers. As an example, in the important paragraph on prostate cancers, the paper by Khoury et al., based on a powerful NGS approach, reports that the detection of HPV sequences with extensive molecular characterization (integration loci etc.) was positive in several types of tumors but that all of the prostate cancer specimens anayzed remained negative. This very strong paper is on the same level as others based on less specific approaches. In other words, if more stringent criteria were used for the analysis of the methodology, the conclusion would be very different and, consequently, the review much more useful for the readers. As a possible approach for instance, this review could highlight HPV detection performed using up to date NGS-based approach, and include anal cancers in their review (may be a review focused on male patients as suggested in the conclusion?). The comparison between the robustness of the data provided by NGS-based investigations on anal carcinoma (Morel, Cancers, 2019) and the paucity of those obtained by the analyses, using similar approaches, of tumors developed in other organs (Khoury for prostate, others most probably exist for bladder, …) should be striking. The authors might thus conclude that HPVs are oncogenic on the anal mucosa (which presents a transition zone) but that the data available on tumors developed in other organs point rather towards passenger infections or mere risk factors of HPVs with no evidence of direct oncogenic effect.

At the era of prophylactic vaccination, it would be important to update the potential individual benefit for boys receiving the HPV vaccine. Of note, there is a strong increase in frequency of head & neck HPV-associated cancers (Lechner, Nature, 2022). A mention of this might also be included, providing a global overview of HPV-associated cancers in males, and reinforcing the quality of the paper in the perspective of vaccine against HPVs in males.

Firstly, I appreciate your kind feedback and suggestions. Following your recommendations, a section discussing the NGS method has been added under the title of HPV detection methods. Subsequently, emphasis has been placed on the strength of the NGS method by including a study conducted by Khoury et al. The changes have been highlighted. Additionally, a section regarding HPV vaccines has been added alongside the sections on anal cancer and penile cancer. In the vaccination section, reference was made to the study by Lechner et al., highlighting the role of HPV in the etiology of head and neck cancers and emphasizing the importance of vaccination regardless of gender.

Round 2

Reviewer 2 Report

Comments and Suggestions for Authors

Thank you for your modifications that add significant information for the readers.

Comments on the Quality of English Language

No specific comment